# Functional analysis of ADARs in planarians supports a bilaterian ancestral role in suppressing double-stranded RNA-response

**Dan Bar Yaacov** [1,2]*

**1** The Shraga Segal Department of Microbiology, Immunology and Genetics, Faculty of Health Sciences, Ben-Gurion University of the Negev, Beer-Sheva, Israel, **2** Department of Integrative Biology, University of Wisconsin-Madison, Madison, Wisconsin, United States of America

* danbary@bgu.ac.il

**Data Availability Statement:** All Sequenced read samples files are available from the Sequence Reads Archive database (accession number PRJNA644394).

## Abstract

ADARs (adenosine deaminases acting on RNA) are known for their adenosine-to-inosine RNA editing activity, and most recently, for their role in preventing aberrant dsRNA-response by activation of dsRNA sensors (*i.e.*, RIG-I-like receptor homologs). However, it is still unclear whether suppressing spurious dsRNA-response represents the ancestral role of ADARs in bilaterians. As a first step to address this question, we identified ADAR1 and ADAR2 homologs in the planarian *Schmidtea mediterranea*, which is evolutionarily distant from canonical lab models (*e.g.*, flies and nematodes). Our results indicate that knockdown of either planarian *adar1* or *adar2* by RNA interference (RNAi) resulted in upregulation of dsRNA-response genes, including three planarian *rig-I-like* receptor (*prlr*) homologs. Furthermore, independent knockdown of *adar1* and *adar2* reduced the number of infected cells with a dsRNA virus, suggesting they suppress a bona fide anti-viral dsRNA-response activity. Knockdown of *adar1* also resulted in lesion formation and animal lethality, thus attesting to its essentiality. Simultaneous knockdown of *adar1* and *prlr1* rescued *adar1(RNAi)*-dependent animal lethality and rescued the dsRNA-response, suggesting that it contributes to the deleterious effect of *adar1* knockdown. Finally, we found that ADAR2, but not ADAR1, mediates mRNA editing in planarians, suggesting at least in part non-redundant activities for planarians ADARs. Our results underline the essential role of ADARs in suppressing activation of harmful dsRNA-response in planarians, thus supporting it as their ancestral role in bilaterians. Our work also set the stage to study further and better understand the regulatory mechanisms governing anti-viral dsRNA-responses from an evolutionary standpoint using planarians as a model.

## Author summary

Today, more than ever, it is crucial to gain a deep understating of our anti-viral defenses. One of the ways to accomplish it is to study the principles governing anti-viral responses across various organisms. ADARs are a group of proteins that act on RNA molecules and alter their sequence compared to the genes that encode them (a process termed RNA

**Funding:** DBY was awarded the Gruss Lipper Post Doctoral Fellowship (http://www.eglcf.org/). DBY was also supported by Ben-Gurion University of the Negev startup grant. The funders had no role in study design, data collection and analysis, decision to publish, or preparation of the manuscript.

**Competing interests:** The authors have declared that no competing interests exist.

editing). In recent years, ADARs have been shown to suppress abnormal anti-viral responses triggered by self-components of the cell (RNA encoded by the cell). Here, we show that the involvement of ADARs in anti-viral response regulation is conserved in planarians (free-living flatworms). We identified two ADAR proteins in planarians and showed that eliminating one (ADAR1) results in animal death and that an anti-viral response commenced in the absence of either ADAR1 or ADAR2. We further identified one of the proteins (PRLR1) that participate in initiating this anti-viral response in planarians, which its mammalian homolog (MDA5) serves a similar role. Thus, our work suggests that ADARs involvement in suppressing aberrant anti-viral response is an ancient evolutionary invention and is likely shared across multicellular organisms with bilateral symmetry.

## Introduction

Adenosine Deaminases Acting on RNA (ADARs) target double-stranded RNA (dsRNA) and introduce adenosine to inosine (A-to-I) changes in RNA sequences [1]. Because inosine is functionally similar to guanosine (G), A-to-I editing can lead to protein recoding, microRNA binding or synthesis changes, or the unwinding of dsRNA [1–3].

ADARs are found across all multicellular animal lineages (including corals) [4] and play several essential roles. For example, vertebrates possess three ADARs: ADAR1 and ADAR2 are catalytically active and known to be essential for viability in mammals [1,2,5–8], while ADAR3 appears catalytically inactive [9]. Mammalian ADAR1 is responsible for most identified editing events, most of which occur in non-coding sequences. For example, in humans, ADAR1 targets mostly inverted *Alu* elements in introns and untranslated regions of mRNA, which form dsRNA structures post-transcriptionally [2]. ADAR2, on the other hand, is thought to mediate its effects primarily through protein recoding [2,8].

While dsRNA molecules are inevitable products of normal cellular function, they are also commonly generated as intermediates of viral replication [10]. As such, they serve as molecular patterns that activate innate immune responses. Organisms must therefore balance between vigilance against foreign dsRNAs without overreacting to innocuous self dsRNA. Emerging evidence suggests a vital role for ADARs in this balancing act. In mammals, for example, ADAR1 is essential to life due to its role in suppressing an interferon (IFN) innate immune response activated by MDA5 (melanoma differentiation-associated protein 5), a dsRNA sensor in the RIG-I (retinoic acid-inducible gene I) like receptor (RLR) family, which binds to long, near-perfectly base-paired structures. [11–13]. Loss of ADAR1 function in mice triggers an embryonically lethal interferon response, which was rescued in *Mda5* knockout mice [11,12]. Similarly, in humans, mutations in both *ADAR1* and *MDA5* (also known as *IFIH1*) are known to cause Aicardi-Goutières syndrome, a devastating inflammatory autoimmune disease [14–16].

Recent studies of the interaction between ADARs and dsRNA-responses in invertebrates have demonstrated intriguing parallels to vertebrates. *Caenorhabditis elegans* encodes two ADARs (ADR-1 and ADR-2) [17,18]. In *adr-1; adr-2* mutant worms, components of the RNA interference (RNAi) pathway–the DICER and ARGONAUTE proteins DCR-1 and RDE-1 – have been shown to process ADAR targets [19]. Additionally, a loss-of-function mutation in *drh1*, which encodes an RLR homolog, suppresses the phenotype of ADAR-deficient worms, an interaction analogous to the observed interaction between ADAR1 and MDA5 in mammals [19,20].

The *Drosophila melanogaster* genome encodes only a single ADAR, which is homologous to mammalian ADAR2 [21]. In flies, Dicer-2, which contains an RNA helicase domain homologous to MDA5 and RIG-I, activates an aberrant anti-viral RNAi response in *Adar* mutants with deficient editing activity [22].

Given these conserved functions between vertebrates and invertebrates, it has been postulated that one of the ancestral roles of ADARs is to prevent aberrant dsRNA-response [2,19,22]. However, the importance of the interaction between ADARs and RLR-mediated or RNAi pathways has only been described in the above invertebrate species. Nematodes and flies represent a limited segment of the animal evolutionary tree–both are members of the superphylum Ecdysozoa–and may lack essential characteristics to inform such evolutionary inferences [23,24]. For example, neither species has an apparent homolog of ADAR1 [17,18], whereas such homologs exist in other invertebrates such as octopuses and oysters (superphylum Spiralia; [25,26]). On the other hand, functional studies of ADARs' role in the dsRNA-response have not yet been conducted in Spiralians.

Therefore, to broaden our perspective on the functional importance of ADARs in dsRNA-response and the evolutionary conservation of this role, we characterized and analyzed ADAR homologs in the planarian *Schmidtea mediterranea*. Along with mollusks, annelids, and several other animal phyla, planarians (free-living platyhelminths) belong to the superphylum Spiralia [27–29]. Planarians are best known for their remarkable ability to regenerate, mediated by a population of pluripotent stem cells (neoblasts) [30]. Interest in their remarkable biology has driven the development of a suite of functional-genetic tools [30–32]. As such, planarians make an attractive, tractable model for molecular-genetic studies from an evolutionary perspective [30].

Here, we describe planarian homologs of ADAR1 and ADAR2 and demonstrate roles for these proteins in the planarian dsRNA-response. RNA interference (RNAi) knockdown of *adar1*, but not *adar2*, resulted in lesions' development and, ultimately, animal death. RNA-Seq analysis of ADAR-knockdown animals demonstrated increased expression of several genes that play roles in anti-viral immunity via RNAi and IFN-like pathways. Significantly, ADAR knockdowns led to a decreased load of *SmedTV*, an endogenous dsRNA virus of *S. mediterranea* [33]. Finally, simultaneous knockdown of *prlr1*, a planarian RIG-I-like receptor, and *adar1* rescued lethality and delayed the dsRNA-response. Collectively, our findings demonstrate the essential immunomodulatory role of the ADAR1 homolog in invertebrates and suggest that this role is evolutionarily conserved across bilaterians.

## Results

### Planarians harbor homologs of human ADAR1 and ADAR2

We identified planarian homologs of human ADAR1 and ADAR2 using reciprocal BLAST between human ADAR1 and ADAR2 and a reference *S. mediterranea* transcriptome [34] as well as phylogenetic analysis (Figs 1A and S1 and S1 Table). In agreement with previous reports, the single *D. melanogaster* Adar grouped with a clade of ADAR2-related proteins, while *C. elegans* ADR-1 was divergent in sequences from other ADARs [6,17,21]. Our phylogenetic analysis clustered ADAR1 together with its canonical homologs. In contrast, ADAR2 showed a high sequence divergence from the canonical ADAR2 homologs and was not assigned to any cluster (S1 Fig). The phylogenetic analysis also supports that ADAR1 and ADAR2 in planarians are divergent from one another (sequence wise), similar to other organisms (S1 Fig). Each planarian *adar* encodes a single RNA-binding domain (RBD) and a deaminase domain with a predicted active site (CHAE motif) (Fig 1A) [18]. The planarian ADAR1 lacks a Z-DNA binding domain, characteristic of canonical ADAR1 homologs (*e.g.*, in

A

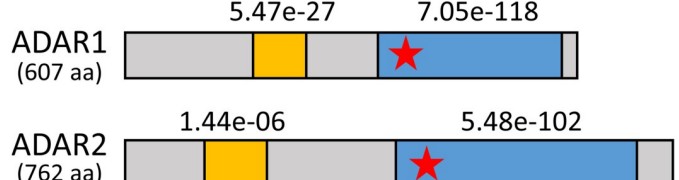

B

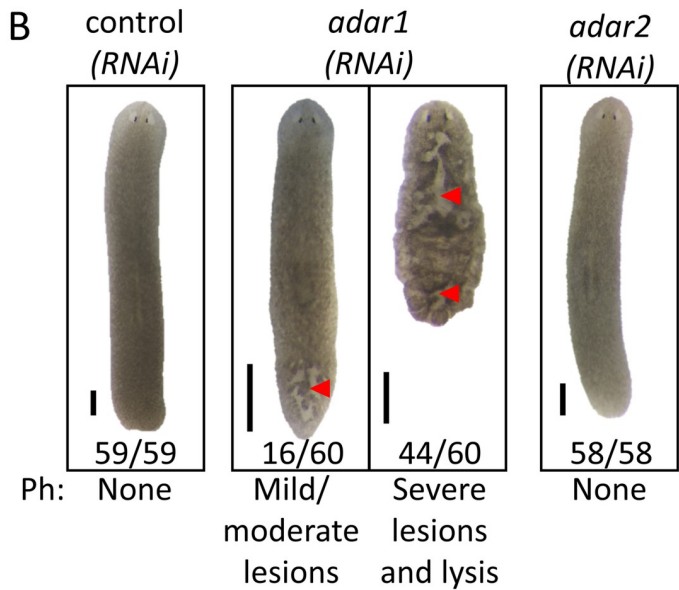

C

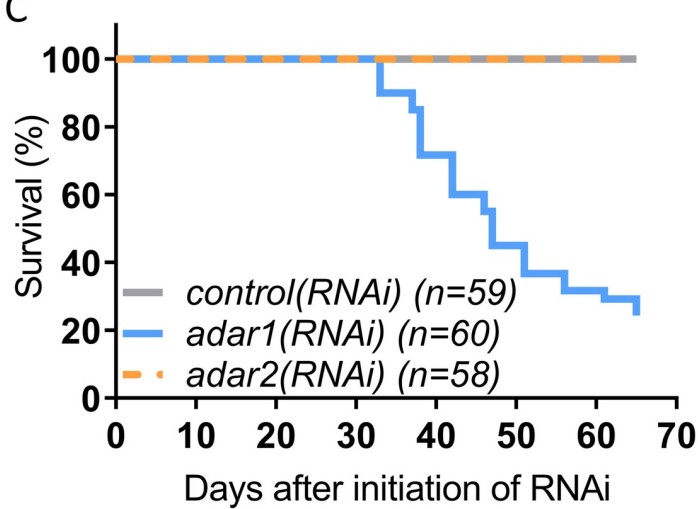

**Fig 1. Planarians harbor homologs of human ADAR1 and ADAR2, and knockdown of *adar1* is lethal.** (A) The domain architecture of ADAR1 and ADAR2 in planarians predicted by NCBI conserved domain search [36]. E-value scores are indicated above the identified domains. aa = amino acids. (B) Knocking down *adar1*, but not *adar2*, results in lesions (red arrowheads), lysis, and lethality. We fed worms dsRNA every 4–5 days (8–12 feedings). N = 5, n ≥ 58, scale bar = 1 mm, Ph = Phenotype. (C) Survival plot of RNAi treated animals from Fig 1B.

humans) [18]. Combined, our analysis indicates that planarians harbor two ADAR homologs, divergent in sequence and domain architecture from ADARs in other systems and one another.

To determine where *adar1* and *adar2* are expressed, we used whole-mount colorimetric *in situ* RNA hybridization (WISH), which revealed a broad expression pattern across the animal body with apparent enrichment in the brain (S2A Fig). Double fluorescent RNA *in situ* hybridization (dbFISH) validated *adar1* and *adar2* broad expression patterns by detecting co-expression with neuronal, neoblast, and gut markers, as well as in surrounding cells (S2B Fig).

## Knockdown of *adar1* is lethal

To examine the function of ADARs in planarians, we used RNAi knockdown of gene expression. RNAi reduced *adar1* and *adar2* transcripts to 22% and 41%-58%, respectively, compared to their levels in control*(RNAi)* animals (S3 Fig and S2 Table). All *adar1(RNAi)* animals were smaller than control *(RNAi)* animals, developed lesions, and 73% (44/60) died (Fig 1B and 1C). In contrast, *adar2(RNAi)* animals did not display any gross morphological phenotype changes, and were similar to control*(RNAi)* animals (Fig 1B and 1C). Notably, the observed phenotype in *adar1(RNAi)* animals did not correspond to the canonical phenotype of neoblast loss (i.e., head regression and ventral curling) [35]. Indeed, *adar1(RNAi)* (and *adar2(RNAi)*) animals were able to regenerate upon the head or tail amputation, performed no more than five days before lesions formed in *adar1(RNAi)* animals (S4 Fig). Furthermore, neither WISH for the pan-neoblast marker *piwi-1* nor flow cytometric analysis of cellular fractions revealed depletion of neoblasts after knockdown of *adar1* or *adar2* (S5 Fig). Therefore, our data collectively suggest that ADAR1 is essential in planarians but that its function is not critical for neoblast maintenance.

## ADAR1 and ADAR2 suppress the expression of genes involved in the dsRNA-response

To elucidate why *adar1* knockdown animals die and to explore possible cellular and molecular effects of *adar2* knockdown, we used RNA sequencing (RNA-Seq) to identify *adar*-dependent gene expression changes after 28 days of RNAi (i.e., before lesion formation in *adar1(RNAi)* animals). RNA-Seq analyses revealed 747 and 448 differentially expressed genes in *adar1 (RNAi)* and *adar2(RNAi)* animals, respectively (False Discovery Rate (FDR) ≤ 0.01; fold change (absolute) ≥ 2; S6A Fig and S2 Table). We identified 345 genes shared between *adar1* and *adar2* RNAi treatments among the differentially expressed genes, suggesting some overlap in function (S6B Fig). Lastly, both *adar1* and *adar2* were among the significantly downregulated genes in RNAi treated animals, with 23% and 41% transcript levels, respectively, as compared to their levels in control*(RNAi)* animals (S2 Table)

Next, we sought to test for over-representation of specific pathways in our differentially expressed gene list. Kyoto Encyclopedia of Genes and Genomes (KEGG) [37] pathway analysis revealed a clear and significant enrichment for upregulated (but not downregulated) genes belonging to multiple anti-viral pathways in both *adar1(RNAi)* and *adar2(RNAi)* animals (S3 Table). Specifically, the RIG-I-like receptor signaling pathway (KEGG:04622) was the most

significantly enriched, in addition to other anti-viral pathways (Fig 2A and S3 Table). We, therefore, hypothesized that in planarians, both ADAR1 and ADAR2 play roles in suppressing defensive responses to dsRNA, similar to their known functions in other animals [11–13,19,22]. Supporting this hypothesis is the finding that other genes that are downstream of RIG-I pathway activation and non-RIG-I pathway genes known to be involved in dsRNA-responses were upregulated (Fig 2B). Along with performing the RNA-Seq experiment after 28 days of RNAi for both *adars*, we also sequenced RNA from worms after 19 days of *adar1* knockdown and control animals. The rationale behind adding this time point was to examine early gene expression changes that preceded the observed *adar1(RNAi)* phenotype. Analyzing this early time point revealed that the above dsRNA-response genes were upregulated in *adar1 (RNAi)* animals as early as 19 days after initiation of RNAi (S2 Table).

As planarian dsRNA-response pathways have not been previously characterized, we focused on seven significantly upregulated genes in our RNA-Seq data as potential indicators of dsRNA-responses (Fig 2C and S2 Table). We focused on a set of representative genes encoding (a) homologs of crucial proteins involved in metazoan dsRNA-responses: RIG-I-like receptors (RLRs), which sense dsRNA (*planarian rig-I-like receptor1 and 3*, *prlr1* and *prlr3*, respectively); (b) Dicer and Argonaute proteins (*dicer1-2*, *ago1* and *ago2-2*), which are core components of the RNA-interference machinery–the primary anti-viral response pathway in invertebrates; and (c) Stat and MX1 proteins (*stat5* and *mx1*), which are associated with interferon- or Jak/Stat-mediated anti-viral functions in vertebrates and invertebrates, respectively (Fig 2B and S2 Table). Quantitative PCR (qPCR) analysis demonstrated that *adar1* knockdown led to more rapid upregulation of dsRNA-response genes than *adar2* knockdown after just ten days of RNAi treatment (Fig 2C). By 19 days, all seven dsRNA-response genes were significantly upregulated in both *adar1(RNAi)* and *adar2(RNAi)* animals (Fig 2C).

Exploring the expression pattern of the seven genes mentioned above using WISH showed a global increase in expression (Fig 2D). Importantly, the observed upregulation of dsRNA-response genes following *adar1* and *adar2* knockdown is not a generic consequence of RNA interference itself, as shown by analyzing RNA-Seq datasets from studies of the effects of RNAi for unrelated genes [38, 39] (Fig 2E). Thus, ADAR1 and ADAR2 likely suppress the expression of dsRNA-response genes in planarians. Furthermore, it is tempting to speculate that the rapid upregulation of dsRNA-response genes could explain why *adar1(RNAi)*, but not *adar2(RNAi)* animals, developed lesions and died.

## ADAR1 and ADAR2 suppress a *bona fide* anti-viral dsRNA-mediated response

We next tested whether increased expression of the dsRNA-response genes following knockdown of either *adar1* or *adar2* constituted a *bona fide* dsRNA-response in planarians. If this were the case, one would expect a negative effect of *adar1* or *adar2* knockdown on RNA viruses in the treated animals (*e.g.*, less infected cells / viral RNA due to upregulation of anti-viral factors). A recent report described a dsRNA virus, *S. mediterranea tricladivirus* (*SmedTV*), in the planarian nervous system [33]. Therefore, we assessed the prevalence *of SmedTV* infected cells and RNA as indicators of the activity of the planarian dsRNA-response. Notably, it was reported that the level of infection (*i.e.*, number of infected cells per worm) varied considerably between individual worms [33]. To overcome this obstacle and obtain sufficient statistical power, we sampled more than 20 worms (pooled from two independent experiments) and counted the number of infected cells in the head of the animals (Fig 3A), as we observed that the majority of SmedTV infected cells, across RNAi treatments, resides in the cephalic ganglion of our sampled animals. Following our prediction, the average number of

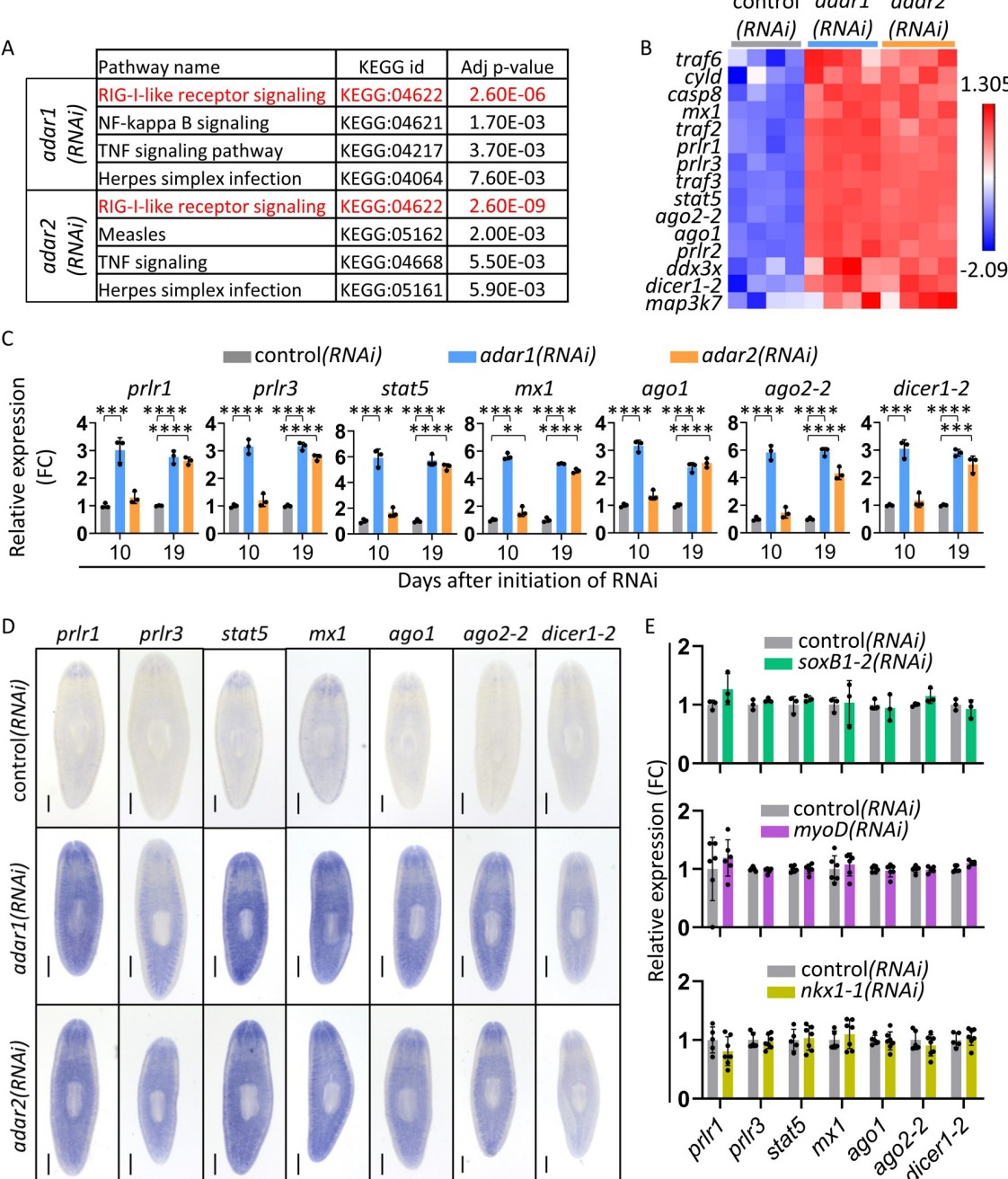

**Fig 2. Knockdown of *adar1* and *adar2* upregulates dsRNA-response genes.** (A) KEGG pathway analysis revealed that RIG-I-like signaling pathway processes are the most enriched in both *adar1(RNAi)* and *adar2(RNAi)* animals after 28 days of RNAi. Here, we show the four most significantly enriched pathways in our RNA-Seq data sets. See S3 Table for all enriched pathways. (B) Heat map illustrating expression of planarian homologs of dsRNA-response genes that are upregulated in both *adar1(RNAi)* and *adar2(RNAi)* animals after 28 days of RNAi (N = 4 (with three animals that were pooled together in each experiment)). FDR ≤ 0.01; Fold change ≥ 2. The expression values used in the gradient color scheme are normalized $\log_2$ CPM values [40]. (C) Relative expression levels (qPCR; mean ± SD; N = 3 (with three animals that were pooled together in each experiment)) of seven dsRNA-response genes in *adar1(RNAi)*, *adar2(RNAi)*, and *control(RNAi)* animals after 10 and 19 days of RNAi. FC = Fold Change. One-way ANOVA with Dunnett's multiple comparisons test for each combination of gene and time point. Adjusted p-value ≤ 0.001 (***) and ≤ 0.0001 (****). (D) Expression patterns of seven dsRNA-response genes by WISH in *adar1(RNAi)*, *adar2(RNAi)*, and *control(RNAi)* animals after 19 days of RNAi support upregulation after knockdown of *adar1* and *adar2*. n ≥ 3 per gene. Scale bar = 500μm. (E) Relative expression levels (mean ± SD; RNA-Seq; N ≥ 3 (with one or more animals that were pooled together in each experiment)) of the seven dsRNA-response genes in *myoD(RNAi)*, *nkx1-1(RNAi)*, and *soxB1(RNAi)* animals and their corresponding controls. FC = Fold

Change. We detected no significant differences by differential gene expression analysis. RNA-Seq data is from previous studies where RNAi was used to knock down the genes mentioned above [38,39]. Genes were knocked down for 23 days (*soxB1*), 49 days (*myoD*), and 63 days (*nkx1-1*).

infected cells was reduced in *adar1(RNAi)* and *adar2(RNAi)* animals (Fig 3A and 3B). The reduction was statistically significant in *adar2(RNAi)* animals ($p < 0.05$) and marginally significant in *adar1(RNAi)* animals ($p = 0.06$). This could be due to the observed large inter-individual variability or could be the result of having a technical outlier (Fig 3A and 3B). In addition to the observed reduction in the number of infected cells, *SmedTV* RNA abundance was also significantly reduced in both *adar1(RNAi)* and *adar2(RNAi)* animals (Fig 3C). Taken together,

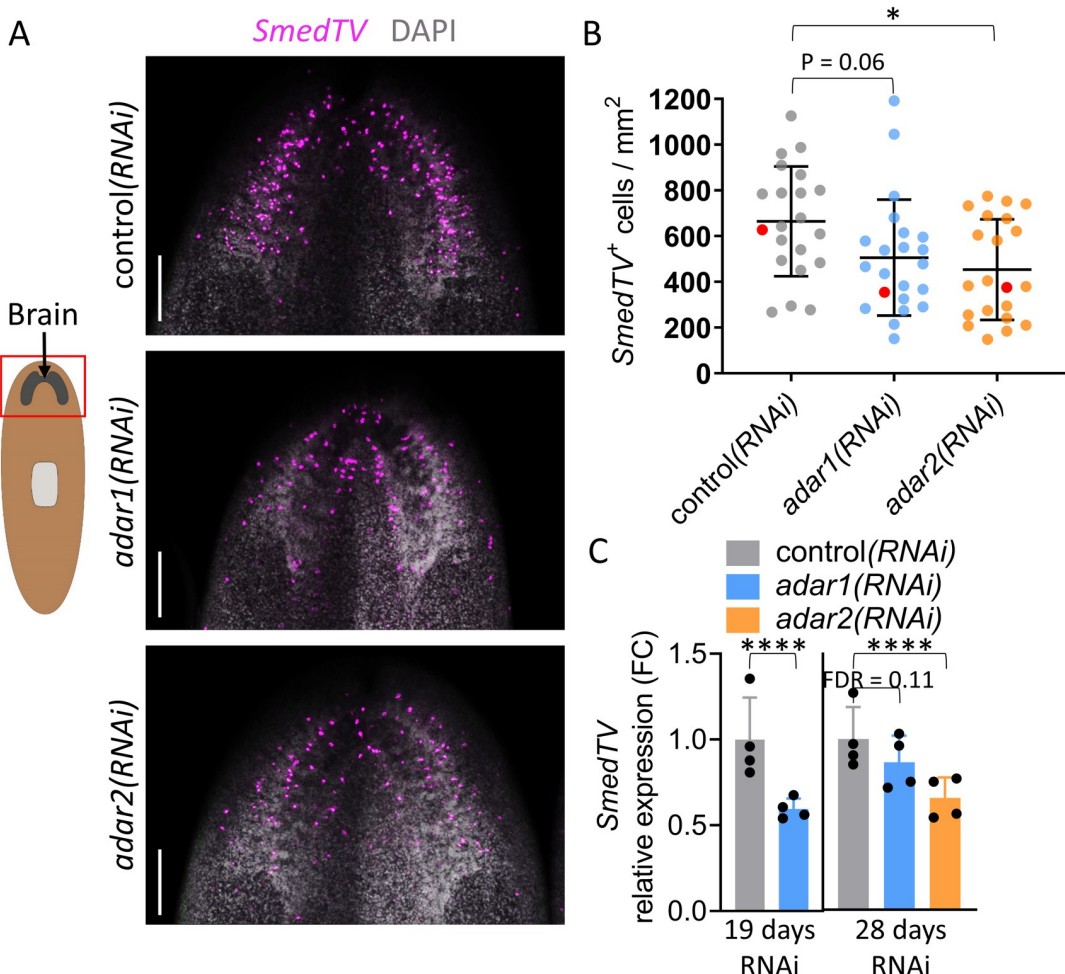

**Fig 3. Knocking down *adar1* or *adar2* reduces viral RNA in infected planarians.** (A) Representative confocal images (FISH–maximum-intensity projection (MIP)) of cells harboring dsRNA of the *S. mediterranea tricladivirus* (*SmedTV*—magenta) in *adar1(RNAi)*, *adar2(RNAi)*, and control*(RNAi)* animals after 21 days. Scale bar = 200 μm. The red box on the cartoon indicates the imaged area. Contrast and brightness were adjusted equally across all three images for better visualization. (B) Quantification of *SmedTV*+ cells in A (mean ± SD; N = 2, n ≥ 20 (pooled animals from both experiments)). One-way ANOVA with Dunnett's multiple comparisons test. Adjusted p-value ≤ 0.05 (*). We pooled the data from two independent experiments after 21 and 23 days of RNAi. Data points corresponding to Fig 3A are marked in red. (C) Relative expression levels (RNA-Seq; mean ± SD; N = 4 (with three animals that were pooled together in each experiment)) of *SmedTV* RNA in *adar1(RNAi)*, *adar2(RNAi)*, and control*(RNAi)* after 19 and 28 days of RNAi. FDR ≤ 0.0001 (****). No RNA-Seq data for *adar2(RNAi)* animals at 19 days of RNAi. FC = Fold Change.

these results suggest that both ADAR1 and ADAR2 dampen the dsRNA-response in planarians.

## PRLR1 is involved in mediating *adar1(RNAi)*-dependent lethality

Next, we hypothesized that the pathologies (lesions and animal death) associated with knockdown of *adar1* stem in part from the relatively rapid upregulation of the dsRNA-response, possibly analogous to an autoimmune response. In mice, knocking out the gene encoding the dsRNA sensor *MDA5* abolishes the IFN-related dsRNA immune response and rescues embryonic lethality in *ADAR1* knockouts [11–13]. We, therefore, asked whether a planarian MDA5 homolog could modulate the ADAR-dependent pathological phenotypes in planarians. We identified three planarian RLR homologs that were upregulated upon *adars* RNAi (S7 Fig and S1 and S2 Tables). Phylogenetic analysis showed that all three diverge in sequence compared to the canonical RLRs (S7A Fig and S1 and S2 Tables) but are closer than Dicer-2 of *D. melanogaster*. In addition, all three contained an N-terminal helicase domain of the DEAD-box helicase superfamily, similarly to canonical RLRs as well as *D. melanogaster* Dicer-2 (S7B Fig). BLAST analysis revealed that planarian PRLR1 displayed the highest homology to human MDA5, a dsRNA sensor (S7C Fig). Therefore, we tested whether *prlr1* knockdown could rescue the planarian lethality caused by *adar1* knockdown (see materials and methods). Indeed, *prlr1* knockdown alleviated lethality in *adar1(RNAi); prlr1(RNAi)* animals, relative to *adar1 (RNAi)* and *adar1(RNAi);* control*(RNAi)* animals (Fig 4A and 4B and 4C). Moreover, lesions started to appear after only four feedings of dsRNA (19 days) in all *adar1* RNAi treatments (single and double RNAi treatments). However, in *adar1(RNAi); prlr1(RNAi)* animals, their severity and frequency decreased (Fig 4A and 4B).

To rule out the possibility that *prlr1* knockdown rescues the *adar1* knockdown defect nonspecifically (for example, by impairing the RNAi pathway itself), we performed a double knockdown of *prlr1* and *piwi-2*. The *piwi-2* gene product is essential for maintaining neoblasts; knockdown of *piwi-2* by itself leads to animal lysis and death [35]. We did not identify any difference in the mortality levels or time of death between *piwi-2(RNAi)*, *piwi-2(RNAi);* control *(RNAi)*, or *piwi-2(RNAi); prlr1(RNAi)* animals (Fig 4D–4F). Therefore, we conclude that PRLR1 function mediates, at least in part, the pathological effects of adar1 knockdown in planarians.

## PRLR1 is involved in mediating dsRNA-response in *adar1(RNAi)* and *adar2(RNAi)* animals

Next, we asked whether PRLR1 function is necessary for the increased dsRNA-response following knockdown of *adar1*. Indeed, *adar1(RNAi); prlr1(RNAi)* animals displayed a lower average expression level of the dsRNA-response genes, relative to both *adar1(RNAi)* and *adar1 (RNAi);* control*(RNAi)* animals after ten days of RNAi (Fig 5A). *adar1* levels did not differ between single and double knockdowns (Figs 5A and S8), further demonstrating that the reduction in expression of dsRNA genes does not result from disruption of *adar1* knockdown but rather from the effect on PRLR1. However, the reduction in expression was transient. After 14 days of RNAi, the expression of all examined dsRNA-response genes was similar between single and double RNAi treatments involving *adar1* (S8 Fig). Thus, it is likely that additional factors are involved in inducing the dsRNA-response in *adar1(RNAi)* animals or that residual amounts of the PRLR1 protein following knockdown of *prlr1* were still able to initiate the dsRNA-response in the absence of ADAR1 (albeit at a lower rate). Next, we asked if PRLR1 plays a role in mediating the dsRNA-response in *adar2(RNAi)* animals. We observed lower average expression levels of all examined dsRNA genes in *adar2(RNAi); prlr1(RNAi)*

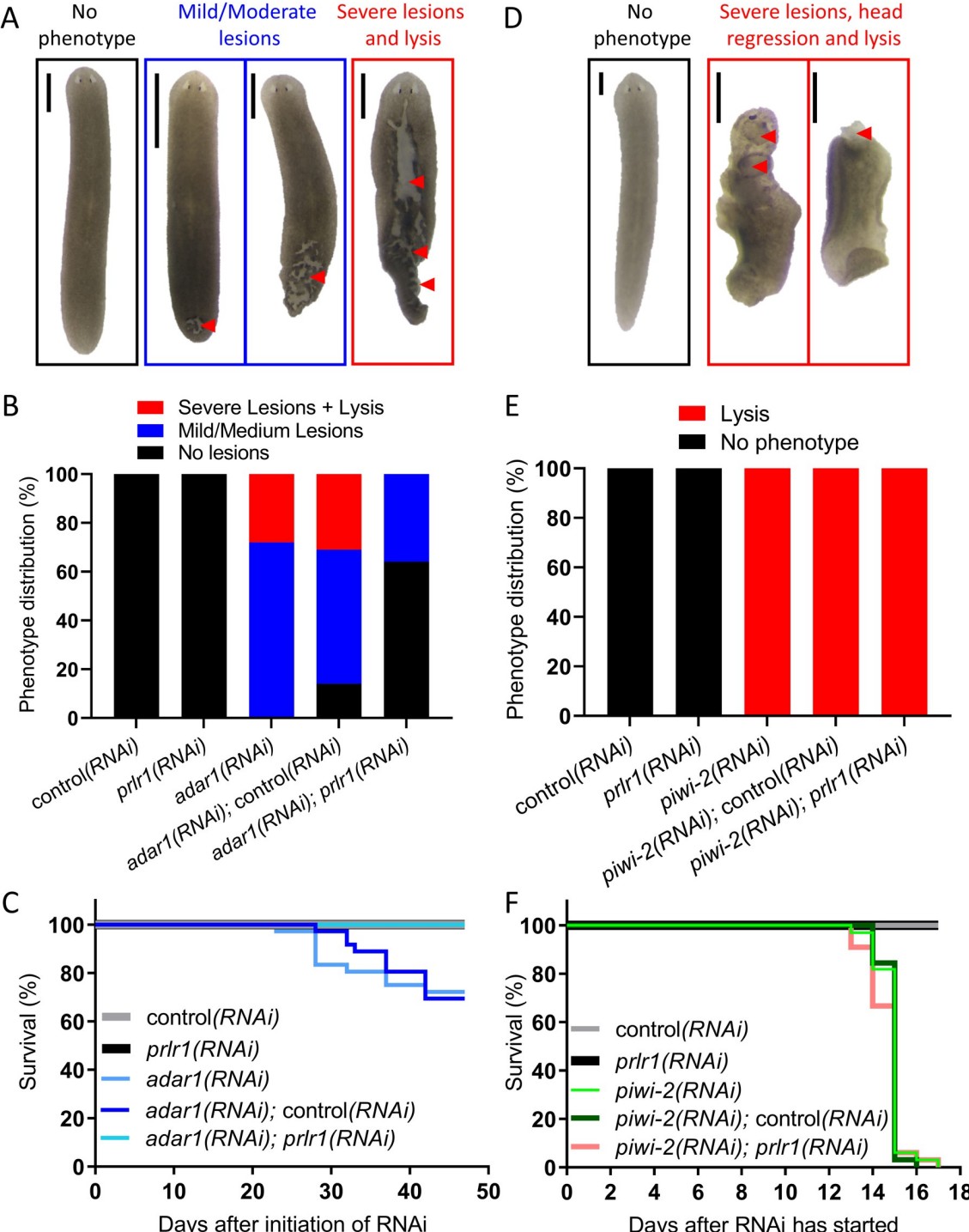

**Fig 4. PRLR1 mediates *adar1* knockdown lethality.** (A) Observed phenotypes in single and double RNAi experiments of *adar1*, *prlr1* and control. Lesions are marked with red arrowheads. (B) Phenotype distribution of A (N = 2, n = 36). Simultaneous knockdown of *adar1* and *prlr1* alleviated the deleterious effect of *adar1* RNAi. (C) Survival plot of the different RNAi treatments shown in B. (D) Observed phenotypes in single and double RNAi experiments of *piwi-2*, *prlr1* and control. Lesions are marked with red arrowheads. (E) Phenotype distribution of D (N = 2, n = 32). Simultaneous knockdown of *piwi-2* and *prlr1* did not alleviate the deleterious effect of *piwi-2* RNAi. (F) Survival plot of the different RNAi treatments shown in E. Scale = 500μm

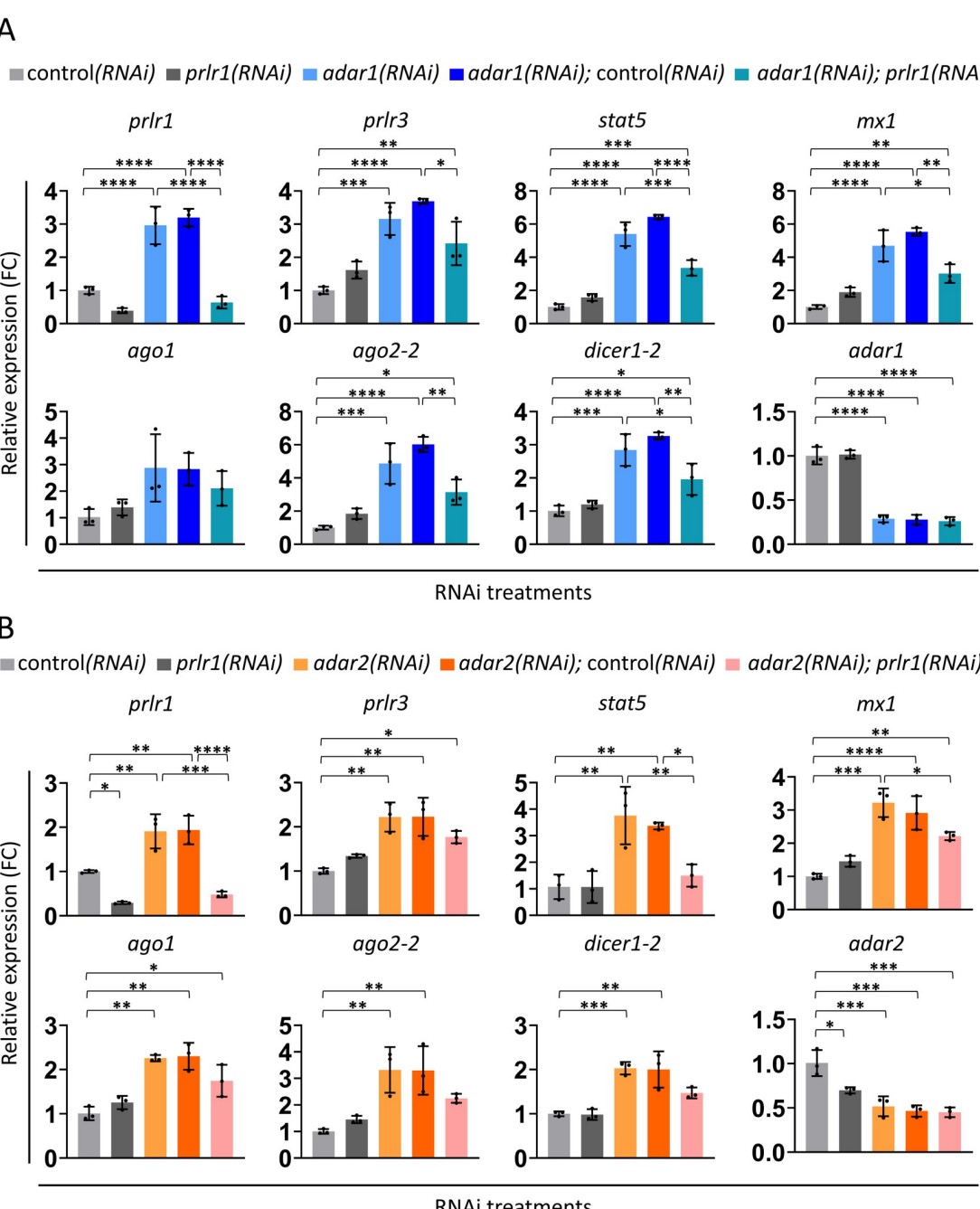

**Fig 5. PRLR1 is involved in mediating dsRNA-response in *adar1(RNAi)* and *adar2(RNAi)* animals.** (A) Relative expression levels (qPCR; mean ± SD; N = 3 (with three animals that were pooled together in each experiment)) of seven dsRNA-response genes and *adar1* after ten days of RNAi. (B) Relative expression levels (qPCR; mean ± SD; N = 3; n = 3) of seven dsRNA-response genes and *adar2* after 14 days of RNAi. FC = Fold change. Statistical analysis—One-way ANOVA with Sidak's multiple comparisons test. Adjusted p-value ≤ 0.05 (*), ≤ 0.01 (**), ≤ 0.001 (***) and ≤ 0.0001 (****).

animals relative to both *adar2(RNAi)* and *adar2(RNAi);* control*(RNAi)* animals after 14 days of RNAi (Fig 5B). However, the effect was not as strong as in the case of *adar1* (*i.e.*, only being statistically significant for *stat5*), raising the possibility of additional factors involved in the regulation of dsRNA-response upon *adar2* knockdown.

In mammals, it was observed that activation of the IFN response by MDA5 in mice with deficient ADAR1 activity leads to an increase in cell death [11,41]. We, therefore, asked whether programmed cell death can explain lesion formation in *adar1(RNAi)* animals. However, we could not detect an increase in programmed cell death (apoptosis) as assayed by TUNEL (S9 Fig) [42], suggesting a different mechanism underlying lesion formation and lysis in *adar1(RNAi)* animals.

Combined, these results are consistent with PRLR1 mediating a dsRNA-response in planarians, which ADARs at least partly suppress in healthy planarians.

## ADAR2 mediates mRNA editing in planarians

ADARs are primarily known for their RNA editing catalytic activity. Furthermore, it has been demonstrated that in mammals, ADAR1 mRNA editing activity at the 3' untranslated regions (UTRs) disrupts base-pairing in endogenous dsRNA structures, which suppresses the autoimmune activation of MDA5 in the cytoplasm [11,12]. Therefore, we analyzed our RNA-Seq datasets for evidence of mRNA editing by ADAR1 and ADAR2 by searching for RNA edits that were present in control*(RNAi)* animals but were absent or at least 50% reduced in *adar1(RNAi)* or *adar2(RNAi)* animals (S10 Fig). This analysis revealed a signature of A-to-I editing (240/246 sites were of A-to-G and T-to-C base changes) attributable to ADAR2, but not ADAR1 (Figs 6A and S11 and S4 Table). Among the 240 ADAR2-dependent edits, 107 events occurred in 51 transcripts with a predicted open reading frame (Fig 6B). Of these, 33.6%, 36.4%, and 30.0% were found in the 5' UTR, coding sequence (CDS), and the 3' UTR regions of the transcripts, respectively (Fig 6B). Within the CDS, 69.2% (27) of the edited sites were also predicted to change amino acid identity, thus possibly affecting protein sequence and function (Fig 6C). In humans, editing events tend to occur in inverted *Alu* repeats [43]. In contrast, except for one site, none of the identified putative edited sites occur in transcripts with homology to known transposable elements. However, the planarian genome is still far from fully annotated, so we cannot exclude editing events in additional planarian-specific transposable elements. Edited sites were found in transcripts expressed in various tissues in planarians and are not limited to a particular type of tissue (Fig 6D). Notably, ADAR1- or ADAR2-dependent putative mRNA editing events were not found in *SmedTV*'s RNA. Thus, the observed effect of *adars(RNAi)* on *SmedTV* RNA and infected cells, is likely editing-independent.

ADARs are known to edit dsRNA structures [2]. Therefore, we analyzed all putative 240 A>G sites, using RNAfold, to detect dsRNA structures across all identified edited transcripts. Our analysis revealed 171 sites that are predicted to pair with a different site in the transcript (S4 Table). Of these, 106 sites are predicted to be embedded in a dsRNA stretch larger than three bases. Since the planarian homolog of ADAR2 contains a dsRNA binding domain (Fig 1A), it is plausible that it also targets dsRNA structures, similar to ADAR proteins in other organisms. Finally, similar to previous reports [2], we did not identify any clear sequence motif around the edited site, except for some enrichment of thymidine that precedes the edited adenosine (Fig 6E). Taken together, according to our results, ADAR2 edits mRNA in planarians, and these edits, in turn, are not essential for planarian viability under standard lab conditions.

## Discussion

ADARs suppress spurious activation of dsRNA-responses to "self" dsRNAs in mammals, flies, and nematodes [11–13,19,20,22]. Thus, it has been suggested that preventing aberrant dsRNA-responses is among the ancestral roles of ADARs.

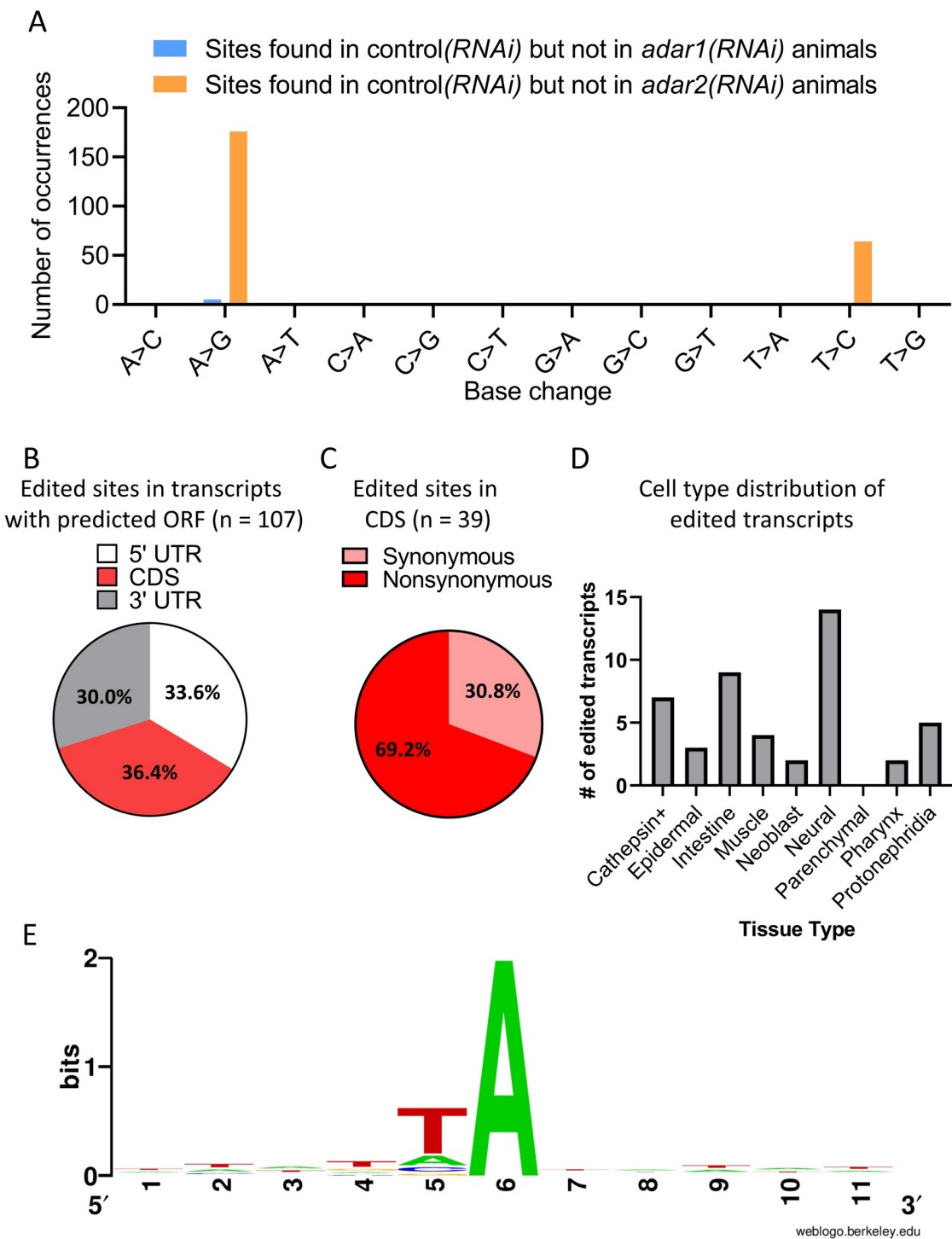

**Fig 6. ADAR2 but not ADAR1 mediates mRNA-editing in *S. mediterranea*.** (A) RNA-Seq analysis reveals hundreds of ADAR2- but not ADAR1- dependent putative A-to-I mRNA editing sites. mRNA-DNA mismatches that were found in all control*(RNAi)* animals but were absent or at least reduced by 50% in all *adar1(RNAi)* or *adar2(RNAi)* animals after 28 days of RNAi are shown (N = 4, n = 3). The analysis revealed ADAR2-dependent enrichment of A-to-G (176 occurrences) and T-to-C (64 occurrences) sites, indicative of A-to-I editing. See S10 Fig for RNA editing discovery pipeline. (B) RNA-editing-site distribution in 51 transcripts with a predicted open reading frame (ORF). (C) The RNA-editing outcome in protein-coding sequences (CDS) for predicted amino acid substitutions (synonymous and nonsynonymous). (D) Cell-type distribution of edited transcripts (n = 31) with detected tissue enrichment in the planarian single-cell RNA-Seq cell atlas [44]. Transcripts can be enriched in more than one tissue. (E) Motif

analysis of five nucleotides upstream and downstream of all 240 ADAR2 putative sites. Some enrichment for thymidine preceding the edited adenosine is observed, but no other well-defined motif.

In support of this hypothesis, we show that both ADAR1 and ADAR2 suppress transcript levels of dsRNA-response genes in planarians. Furthermore, we observed a reduction in *SmedTV*<sup>+</sup> infected cells and RNA upon knockdown of *adar1 or adar2*, which suggests that both ADARs suppress a *bona fide* dsRNA anti-viral response (Fig 3). The ability of *prlr1 (RNAi)* to rescue *adar1(RNAi)*-dependent lethality in planarians and to affect the induction of the dsRNA-response suggests that both are causally linked.

If the dsRNA-response is harmful, why do *adar1(RNAi)* but not *adar2(RNAi)* animals develop lesions and die? One explanation is that *adar2* was not knocked down sufficiently (S3 Fig). However, three key findings does not support this explanation: 1. The expression levels of dsRNA-response genes were comparable after 19 days of RNAi for both *adar*s (Fig 2); 2. Our RNA-seq data that is derived from four independent experiments, and is more accurate than our qPCR analysis, detected that the expression of adar2 was reduced to ~40% of its levels in control(RNAi) animals, yet no phenotype was observed in any of these biological replicates (S2 Table); 3. knocking down adar2 was sufficient to eliminate hundreds of putative editing events (Fig 6), thus attesting to the loss of the enzymatic activity of ADAR2, which indicate that the knockdown was effective. An alternative explanation could be that the rapid induction of the dsRNA-response may be sufficient to induce lesion formation and lysis in *adar1(RNAi)* animals. In addition, *adar1* knockdown has a more significant effect on gene expression than *adar2* knockdown (*i.e.*, the expression of more genes is affected; S6 Fig and S2 Table). Therefore, it is plausible that ADAR1 has additional roles impaired by knockdown when combined with the induction of the dsRNA-response apply cumulative stress that leads to lethality in *adar1(RNAi)* animals.

Previous work has suggested that the editing activity of ADARs is required for preventing dsRNA sensing and activation of spurious dsRNA-responses [11–13,19,20,22]. Indeed, we identified hundreds of putative editing events by ADAR2. However, depletion of ADAR2-dependent RNA editing did not affect the viability of the animals under laboratory conditions. Thus, RNA editing by ADAR2 is possibly not essential for viability, which does not preclude its importance in certain environmental conditions. In contrast to ADAR2, we did not detect ADAR1-dependent edits. Thus, ADAR1 either does not have mRNA editing activity (despite having a predicted active site), its edits are restricted to the non-polyadenylated fraction of the transcriptome, or ADAR2 can compensate for the loss of ADAR1 upon knockdown, which should be assessed in the future. Finally, we could not detect putative mRNA editing events in the RNA of SmedTV. Thus, the effect of ADARs on the abundance of SmedTV's RNA and infected cells is likely indirect.

In mice, ADAR1 is responsible for most editing events, especially in non-coding regions, while ADAR2 mainly edits coding regions [2]. Knocking out each *Adar* gene individually leads to a lethal phenotype [1,11–13]. Thus, at least in part, the activities of ADAR1 and ADAR2 are not redundant. Since *D. melanogaster* and *C. elegans* harbor only a single *ADAR* gene [21,45], investigating the redundancy between ADAR homologs in these model systems is impractical. Planarians, however, harbor two ADAR homologs. Our results indicate both similarities and differences upon *adar1* and *adar2* knockdown in planaria. Thus, planarians could serve as an attractive model for investigating the interaction between ADAR paralogues. Finally, we and others [19,22], showed that ADAR2 orthologue could induce a dsRNA response phenotype, it may be of interest to examine its involvement in anti-viral dsRNA response also in mammals.

Our study sets the stage to elucidate further the regulation of dsRNA-response in planarians, which represent evolutionarily distinct bilaterian (superphylum Spiralia) from other widely used invertebrate models (*e.g.*, nematodes and arthropods). In invertebrates, the RNAi system is thought to execute the lion's share of the anti-viral immune response [46–50]. Nevertheless, the RNAi pathway is not the only anti-viral response in invertebrates. For example, it has been shown that in mosquitos, the JAK-STAT pathway plays a role in fighting viral infections, which is similar to the vertebrate interferon system [46,51,52]. However, evidence for non-RNAi anti-viral dsRNA immune responses is poorly documented for invertebrates other than insects. Thus, future research in planarians could help uncover conserved/novel elements in the dsRNA-response pathway. For example, we identified a planarian homolog for STAT, a transcriptional mediator of the interferon response in mammals (Fig 2). Thus, it is possible that JAK-STAT signaling is involved in mediating the downstream PRLR1-dependent dsRNA upregulation in planarians. If this is the case, it will be interesting to test whether secreted factors, analogous to interferons in mammals, play a role downstream of PRLR1-mediated upregulation. In theory, perturbing the expression of key regulators in the dsRNA-response pathway should, in turn, interfere with the *adar1* knockdown phenotype or prevent upregulation of dsRNA-response genes. Thus, *adar1* knockdown could be used as a tool to elucidate further different factors that are involved in planarians dsRNA-response.

In conclusion, our work supports deep evolutionary functional conservation of ADARs in suppressing aberrant dsRNA-responses initiated by RIG-I-like receptor homologs. In addition, it sets the stage to study further and better understand the regulatory mechanisms governing anti-viral dsRNA-responses from an evolutionary standpoint, using planarians as a model.

## Material and methods

### Planarian husbandry

Planarians from the asexual strain CIW4 [53] were kept in 1x Montjuïc water (1.6 mM NaCl, 1.0 mM CaCl$_2$, 1.0 mM MgSO$_4$, 0.1 mM MgCl$_2$, 0.1 mM KCl and 1.2 mM NaHCO$_3$ in Milli-Q water, pH 6.9–8.1) supplemented with 50 µg/mL gentamicin (Gemini Bio-Products # 400–108) [54]. Worms were kept in unsealed Ziploc containers or 100mm Petri dishes. Worms were kept in unsealed Ziploc containers or 100mm Petri dishes. We irradiated worms on the top shelf of a benchtop X-ray irradiator (CellRad, Precision X-ray) with 60 Gray at 130 kV, 5 mA to ablate stem cells.

### Identification of *adar1* and *adar2* homologs

I used tBLASTn with human ADAR1 and ADAR2 protein sequences to find the planarian homologs in the Dresden version 6 transcriptome (dd_v6) in PlanMine [34] (S1 Table). We then used BLASTx to query these transcripts against the human RefSeq proteome to confirm that they are the closest homologs to the human proteins (S1 Table). In order to identify conserved motifs/domains, we used NCBI's conserved domain search [36].

### Phylogenetic analysis

To construct a Maximum Likelihood tree, we identified (BLASTp) homologs to human ADARs in representative members of the different animal taxa (S1 Fig and S1 Table). The evolutionary history was inferred by using the Maximum Likelihood method based on the JTT matrix-based model [55]. The bootstrap consensus tree inferred from 1000 replicates is taken to represent the evolutionary history of the taxa analyzed [56]. Branches corresponding to partitions reproduced in less than 50% bootstrap replicates are collapsed. The percentage of

replicate trees in which the associated taxa clustered together in the bootstrap test (1000 replicates) are shown next to the branches [56]. Initial tree(s) for the heuristic search were obtained automatically by applying Neighbor-Join and BioNJ algorithms to a matrix of pairwise distances estimated using the JTT model and then selecting the topology with a superior log-likelihood value. This analysis involved 27 amino acid sequences. There were a total of 1526 positions in the final dataset. Evolutionary analyses were conducted in MEGA X [57]. The same strategy was used to infer the phylogenic relationships between RLR homologs (S7 Fig). The analysis involved 19 amino acid sequences (S1 Table). There was a total of 1764 positions in the final dataset.

## Synthesis of dsRNA and riboprobes

I synthesized cDNA using the iScript kit (Bio-Rad, #1708890). For each gene of interest, we amplified from cDNA a 222–1557 bp fragment (S5 Table and S6 Table). PCR products were visualized on 1% agarose gel in TAE buffer, cleaned, and concentrated using the DNA Clean & Concentrator-5 kit (Zymo Research #D4004). Cleaned PCR products were cloned into pJC53.2, a vector designed to allow TA-cloning and subsequent production of riboprobes or dsRNA [58]. Plasmids with cloned genes served as a template for PCR amplification using a T7 primer (S5 Table). PCR products were cleaned as described above and incubated with T7 RNA polymerase (Newmark Lab) to synthesize dsRNA [59]. To produce antisense riboprobes, cleaned PCR products were incubated with SP6 or T3 RNA polymerases (Newmark Lab) as previously described [31,58].

## RNA interference

In order to knock down gene expression, dsRNA (>1μg/μL) was mixed with bovine liver puree in a 1:4 ratio. Worms were starved for 5–14 days before the initiation of experiments. Worms (10–40) were placed in 100x25 mm plates (Fisher Scientific #FB0875711) containing 60mL of Montjuïc water supplemented with 50 μg/mL gentamicin (Gemini Bio-Products # 400–108) and were fed between 50–100μL of liver/dsRNA mixture for 2–4 hours. Worms were then moved to new plates containing fresh media. Feeding occurred every 4–5 days. We used dsRNA synthesized from stock pJC53.2 plasmid for control RNAi, encoding the *ccdB* and *camR* bacterial genes, which are not encoded by the planarian genome. For double RNAi experiments, animals were fed four days of *prlr1* dsRNA (in *prlr1(RNAi)*, *adar1(RNAi); prlr1 (RNAi)* and *adar2(RNAi); prlr1(RNAi)* animals) or control dsRNA (in all other treatments), followed by dsRNA treatments as indicated in the graphs (Figs 4 and 5 and S8). Importantly, *adar1(RNAi)* and *adar1(RNAi);* control*(RNAi)* animals displayed reduced feeding activity after 28 days of RNAi (six feedings of dsRNA). Therefore, to control for the possible impact of feeding behavior on knockdown efficiency, feeding was stopped after 28 days of treatment. This likely accounts for the reduction in lethality observed in Fig 4 compared to the original *adars* RNAi experiments, where animals were treated for 38–52 days (8–12 feeding cycles) (Fig 1).

## DNA extraction and sequencing

Before DNA extraction, worms were treated with 7.5% (wt/vol) N-acetyl cysteine in PBS for 10 minutes, followed by PBS-only wash for 5 minutes. DNA was extracted using the Gentra Puregene Tissue Kit (Qiagen, #158667). A TruSeq Nano DNA LT library (Illumina, 125bp, paired-end) was constructed and sequenced at the UW-Madison Biotechnology center on a HiSeq 2500 platform.

## RNA extraction, sequencing, and analysis

According to manufacturer instructions, we used TRIzol Reagent (Invitrogen #15596026) to lyse and extract RNA from intact worms (N = 4, n = 3). RNA was DNase-treated (New England Biolabs #M0303S) and cleaned using RNA Clean & Concentrator-5 kit (Zymo Research #R1013). TruSeq Stranded mRNA libraries (Illumina, 100bp, paired-end) were constructed and sequenced at UW-Madison Biotechnology center on a NovaSeq 6000 platform. CLC Genomics Workbench (Qiagen) was used to map the reads to the dd_v6 transcriptome and to identify differentially expressed genes between *adar1(RNAi)* or *adar2(RNAi)* and control*(RNAi)* animals. BLASTx determined homology of differentially expressed genes to the RefSeq database of *H. sapiens*, *C. elegans*, and *D. melanogaster*. The BLAST hit with the lowest e-value is shown in S2 Table.

Previous work identified homologs of the planarian *Dugesia japonica* for *dicer1* and *ago2* [60] that are also found in the transcriptome of *S. mediterranea* (S6 Table) but do not correspond to the identified transcripts in our analysis. Therefore, we named the homologs we identified as *smed-dicer1-2* and *smed-ago2-2*.

Sequenced read samples have been deposited in Sequence Reads Archive (SRA accession–PRJNA644394). We also analyzed RNA-Seq expression data for *soxB1* RNAi, *myoD* RNAi, *nkx1-1* RNAi and controlled RNAi in SRA accessions SRP158958 [38] and SRP107206 [39].

## KEGG pathway analysis

In order to detect the enrichment of known pathways in our set of differentially expressed genes, we used KEGG pathway analysis [37]. Specifically, g: Profiler (https://biit.cs.ut.ee/gprofiler/gost) [61] and DAVID (https://david.ncifcrf.gov/) [62] were used independently to perform the KEGG pathway analysis (S3 Table). All transcripts with a BLAST hit to a human homolog were used (see "NCBI accessions" tab in S3 Table). Only pathways considered significantly enriched with an adjusted P-value below 0.05 are reported.

## RNA editing analysis

In order to detect RNA editing sites, our sequenced genomic DNA reads were mapped to the dd_v6 transcriptome, and a consensus genomic-DNA-based sequence corresponding to each transcript was extracted. RNA reads from control*(RNAi)*, *adar1(RNAi)*, and *adar2(RNAi)* samples (four biological replicates) along with the genomic DNA reads, were mapped to the DNA-based consensus transcriptome. Only reads with at least 80% identity and at least 80% of their lengths matched the reference sequence. To identify edited sites in transcripts, we first excluded mismatches between our RNA sequences and the DNA-based consensus transcriptome present in our DNA-Seq reads (sites with a variant frequency above 0.5%). Second, we kept only mismatches with a frequency of at least 2% in the control animals. All identified sites had sequenced-read coverage $\geq$ 10, with four unique reads supporting putative RNA editing events. Third, only mismatches that were found in all four control*(RNAi)* samples but were absent or reduced in frequency by at least 50% in *adar1(RNAi)* or *adar2(RNAi)* animals were called. Notably, we allowed the detection of editing events in *adar1(RNAi)*, *adar2(RNAi)* and DNA samples with coverage of only four reads, to exclude false positive sites identified due to lack of data in these samples. Sanger sequencing of cDNA from WT and RNAi worms was used to validate selected RNA editing sites with high editing levels (that allow reliable editing detection with Sanger sequencing).

## RNA secondary structure prediction

In order to examine the RNA secondary structure, we used a locally installed RNAfold program (RNAfold ViennaRNA-2.5.0) to predict the RNA secondary structure with minimum free energy. We analyzed the entire set of edited transcripts, and examined whether edited sites could pair and what is the length of a detected dsRNA structure, providing that at least 80% of the bases are paired.

## Motif analysis

We used weblogo [63] at http://weblogo.berkeley.edu/logo.cgi to identify the possible motif of ADAR2-dependent putative sites. Five nucleotides upstream and downstream to the edited site were examined.

## qPCR

I used the GoTaq master mix (Promega, #A6002) on a StepOnePlus real-time PCR machine and software (Applied Biosystems) to measure the expression levels of specific genes. Endogenous expression levels of all genes were normalized to *β-tubulin* as previously described [64]. Each experiment included three technical replicates for each of three biological replicates per treatment. All primers can be found in S5 Table.

## *In situ* RNA hybridization

As previously described, colorimetric In situ hybridization (ISH) and fluorescent In situ hybridization (FISH) experiments were performed [31]. Specifically, 10–40 starved (at least four days) worms were killed and stripped of mucus by incubating them for 10 minutes in 7.5% (wt/vol) N-acetyl-L-cysteine (NAC) dissolved in PBS, followed by fixation in 4% (wt/vol) formaldehyde (Sigma-Aldrich #252549) in PBSTx (PBS + 0.3% Triton X-100, Fisher BioReagents, #BP151-500). Worms were stored in 100% methanol at -30°C for a minimum of 16h. Worms were bleached for 3 hours to overnight in formamide-containing solution under bright light, followed by incubation in a proteinase K solution (5 μg/mL proteinase K + 0.1% SDS in PBSTx). For colorimetric ISH, we used digoxigenin-containing (DIG) antisense probes in combination with anti-DIG-AP (alkaline phosphatase) antibody (1:2000, Millipore-Sigma #11093274910). For FISH we used DIG and/or DNP (dinitrophenol) containing probes, detected by tyramide signal amplification using anti-DIG-POD (peroxidase) (1:2000, Millipore-Sigma #11207733910) or anti-DNP-HRP (horseradish peroxidase) (1:2000, Vector laboratories #MB-0603). All samples in each experiment were processed in the same way in a side-by-side manner.

## TUNEL

TUNEL was performed as previously described [42] with the following modifications. Worms were killed, fixed, and formamide-bleached as described in the *in situ* hybridization section. Worms were then incubated for four hours at 37°C in 20 μL of TdT reaction mix (0.8 μM DIG-dUTP, 39.2 μM dATP, 1× reaction buffer (New England Biolabs #M0315L), 250 μM CoCl2 (New England Biolabs #M0315L), 0.5 units/μL terminal transferase (New England Biolabs #M0315L)–final concentrations). Worms were then washed, blocked, and incubated overnight with anti-DIG-POD (peroxidase) (1:2000, Millipore-Sigma #11207733910) as described in the *in situ* hybridization section.

## Imaging

We used a Leica M80 stereomicroscope to image live worms using an iPhone 6 camera mounted on a microscope adapter (iDu LabCam #B00O98AHH0). Whole-mount ISHs (WISH) were imaged on a Zeiss AXIO Zoom V16. WISH images were processed using Photoshop (Adobe) or Gimp for white background adjustment and image cropping. In S2A Fig, color curves were adjusted equally across all three images to visualize expression patterns better. Fluorescence images (immunofluorescence and FISH) were captured using a Zeiss LSM 880 confocal microscope and either a 20X (Plan-Apochromat 20x/0.8) or a 63X objective (Plan-Apochromat 63x/1.4). Zen software (Zeiss) was used for these experiments. For comparisons of different treatments, we used the same settings for image collection. Cell counts were normalized to the imaged area. TUNEL-positive cells were counted manually using imageJ [65].

## Statistical analysis

GraphPad PRISM 8.2 was used for all statistical analysis, except for the differential expression analysis, which was conducted using CLC Genome Workbench (Qiagen) as described above.

## Flow cytometry

Starved (5 days) RNAi-treated worms were dissociated and analyzed by flow cytometry as previously described [35,66]. Briefly, worms from each RNAi treatment (n = 8) were cut to small pieces and dissociated for 25 minutes in CMF buffer (0.1 mg/mL sodium phosphate monobasic monohydrate, 0.2 mg/mL sodium chloride, 0.3 mg/mL potassium chloride, 0.2 mg/mL sodium bicarbonate, 10 mg/mL BSA, 0.02 M HEPES, 0.02 M glucose, 50 µg/mL gentamicin sulfate in ultra-pure water) and collagenase (final concentration 1 mg/mL). We used 100 µm, 40 µm, and 20 µm sieves to remove large pieces of un-dissociated tissue. Cells were stained for 90 minutes in 500 µL CMF buffer with Hoechst 33342 (20 µg/ml) and calcein-AM (0.05 µg/mL). Cells were centrifuged at 310 g to remove unincorporated calcein and 500 µL of CMF buffer with Hoechst 33342 (20 µg/mL) and propidium iodide (1 µg/mL) were added before flow cytometry. Flow cytometry was conducted on a BD FACS Aria II BSL-2 Cell Sorter at the flow cytometry lab at the University of Wisconsin Carbone Cancer Center. Cytometric data were analyzed and visualized using FlowJo 10 (https://www.flowjo.com).

## Supporting information

**S1 Fig. Phylogenetic analysis clusters the planarian ADAR1 and ADAR2 with canonical homologs.** A maximum-likelihood phylogenetic tree of ADAR, ADAD (adenosine deaminase containing domain, also known as TENR), and ADAT (adenosine deaminase acting on tRNAs) homologs, with species representing different bilaterian lineages and cnidarians, places planarian ADAR1 together with its canonical homologs while revealing a high level of divergence in planarian ADAR2. Bootstrap values (percentages based on 1000 replicates) are indicated at the base of the branches. Branches corresponding to partitions reproduced in less than 50% bootstrap replicates are collapsed.
(TIF)

**S2 Fig. *adar1* and *adar2* are broadly expressed.** (A) Expression patterns of *adar1* and *adar2* by WISH (n ≥ 4). Black arrowheads mark enriched expression in the cephalic ganglion. The neoblasts markers *soxP-1* and *soxP-2* are used to control for probe specificity. Scale bar = 500µm. (B) Representative confocal images of dbFISH show co-expression of *adar1* and *adar2* (shown in magenta) with neuronal, gut, and neoblast markers (*pc2*, *mat*, and *gH4*,

respectively, shown in green). Maximum-intensity projection of a 4 μm section. Scale bar = 20μm.
(TIF)

**S3 Fig. Validation of *adar1* and *adar2* knockdown efficiency by RNAi.** Relative expression levels (qPCR; N = 3 (with three animals that were pooled together in each experiment); mean ± SD) of *adar1* (left) and *adar2* (right) in *adar1(RNAi)*, *adar2(RNAi)*, and control *(RNAi)* after 19 days of RNAi. FC = Fold Change. Statistical comparisons are based on one-way ANOVA with Dunnett's multiple comparisons test (each treatment compared to control). Adjusted p-value ≤ 0.01 (**) and ≤ 0.0001 (****).
(TIF)

**S4 Fig. Knocking down a*dar1* and *adar2* does not block regeneration.** Head and tail regeneration 14 days post-amputation. The red dotted line represents the amputation plane. Worms were fed dsRNA every 4–5 days. n = 20 from two independent experiments (10 worms each) after 19 and 23 days of RNAi (four and five feedings, respectively). Scale bar = 1mm.
(TIF)

**S5 Fig. Neoblasts are present after *adar1* and *adar2* RNAi.** (A) Expression of *piwi-1*, a pan-neoblast marker, by WISH shows no stem-cell depletion in *adar1(RNAi)* or *adar2(RNAi)* animals. Scale bar = 500μm. (B) Cytometry plots quantifying stem cells (neoblasts) (X1 and X2 gates) and post-proliferative cells (Xins) show no stem-cell depletion in *adar1(RNAi)* and *adar2(RNAi)* animals after 28 days of RNAi (n = 8). X-irradiated worms served as a positive control for stem-cell loss and gating (60 Gy, 48 hours post-irradiation).
(TIF)

**S6 Fig. RNA-Seq reveals hundreds of differentially regulated genes in *adar1(RNAi)* and *adar2(RNAi)* animals.** (A) Left—Heat map of 356 upregulated and 391 downregulated genes after 28 days in *adar1(RNAi)* animals. Right—Heat map of 289 upregulated and 159 downregulated genes after 28 days in *adar2(RNAi)* animals. N = 4 (with three animals that were pooled together in each experiment); FDR ≤ 0.01; Absolute fold change ≥ 2. The expression values used in the gradient color scheme are normalized $\log_2$ CPM values [40]. (B) Venn diagram shows an overlap of differentially regulated genes in *adar1(RNAi)* and *adar2(RNAi)* animals.
(TIF)

**S7 Fig. Phylogenetic analysis reveals sequence divergence between the planarian RLRs homologs and canonical (vertebrates) RLRs.** (A) A protein maximum likelihood phylogenetic tree with species representing different bilaterian lineages and cnidarians demonstrates that *S. mediterranea* RLR homologs are distinct from canonical RIG-I-like receptors (RIG-I, PRLR1, and LGP2). *D. melanogaster* Dicer-2 served as an outgroup as it harbors a helicase domain homologous to canonical RLRs. Bootstrap values (1000 replicates) are indicated at the base of the branches. Branches corresponding to partitions reproduced in less than 50% bootstrap replicates are collapsed. (B) Domain architecture of PRLR1—PRLR3 as predicted by NCBI conserved domain search [36]. E-value scores are indicated next to the identified domains. aa = amino acids. (C) BLASTX analysis of the three planarian transcripts encoding RLR homologs against the protein sequence of MDA5 (human).
(TIF)

**S8 Fig. PRLR1 mediates *adar1*-dependent upregulation of dsRNA-response transiently.** Relative expression levels (qPCR; mean ± SD; N = 3 (with three animals that were pooled together in each experiment)) of seven dsRNA-response genes and *adar1* after 14 days of RNAi. FC = Fold change. Statistical analysis—One-way ANOVA with Sidak's multiple

comparisons test. Adjusted p-value $\leq 0.05$ (*), $\leq 0.01$ (**), $\leq 0.001$ (***) and $\leq 0.0001$ (****).
(TIF)

**S9 Fig. Knocking down *adar1* or *adar2* does not induce apoptosis.** Confocal images (FISH–single plane) and quantification of TUNEL staining after 23 days of *adar1* or *adar2* RNAi. The dashed black square represents the region corresponding to the images shown in the cartoon, while the red square represents the imaged and quantified area. Scale bar = 50 μm. One-way ANOVA with Dunnett's multiple comparisons test (each treatment compared to control). No significant differences were detected.
(TIF)

**S10 Fig. RNA editing analysis pipeline.** See also the materials and methods section.
(TIF)

**S11 Fig. Sanger sequencing validation of putative RNA editing sites.** Sanger sequencing validates 8/9 putative A-to-G mismatches identified in our RNA-Seq analysis between *adar2 (RNAi)* and control*(RNAi)* samples. Here, genomic DNA (WT) and cDNA from *adar2(RNAi)* animals harbor adenosine in these sites, while cDNA from control*(RNAi)* animals contain guanosine or mixed guanosine and adenosine (indicative of A-to-I editing). Red boxes denote the validated sites.
(TIF)

**S1 Table. Identification of *adar1* and *adar2* homologs in planarians and sequences used for phylogenetic analysis.** This table contains three sheets: "tBLASTn Human vs. planarian" contains a tBLASTn analysis of human ADAR1 and ADAR2 protein sequences against the planarian transcriptome (dd_Smed_v6); "BLASTx planarian vs. Human" contains a BLASTx analysis of the identified planarian sequences against the human RefSeq protein database; "S1 Fig" contains the protein sequences used to construct the phylogenetic tree in S1 and S5 Figs contains the protein sequences used to construct the phylogenetic tree in S5 Fig
(XLSX)

**S2 Table. RNA-Seq differential expression analysis.** This table contains six sheets. Each sheet contains the identified differentially expressed genes in *adar1(RNAi)* or *adar2(RNAi)* compared to control*(RNAi)* animals. UpReg–Upregulated; DownReg–Downregulated.
(XLSX)

**S3 Table. KEGG pathway analysis.** This table contains five sheets–"NCBI accessions" contains all protein accession numbers (RefSeq) from Table S2 for *adar1(RNAi)* and *adar2(RNAi)* animals (28 days of RNAi); "gProfiler_Upregulated genes" contains the identified enriched pathways in *adar1(RNAi)* and *adar2(RNAi)* animals according to gProfiler; "gProfiler_Downregulated genes" contains the identified enriched pathways in *adar1(RNAi)* and *adar2(RNAi)* animals according to gProfiler; "DAVID_Upregulated genes" contains the identified enriched pathways in *adar1(RNAi)* and *adar2(RNAi)* animals according to DAVID; "DAVID_Downregulated genes" contains the identified enriched pathways in *adar1(RNAi)* and *adar2(RNAi)* animals according to DAVID.
(XLSX)

**S4 Table. RNA editing analysis.** This table shows RNA-DNA mismatches that were found in all four control*(RNAi)* animals but are absent or reduced in by least 50% in all four *adar1 (RNAi)* or *adar2(RNAi)* animals.
(XLSX)

**S5 Table. Primers used in this study.** This table contain the primer names used in this study, their dd_v6 accession numbers, sequence, length and use.
(XLSX)

**S6 Table. Genes mentioned in this study.** This table contains gene names of genes mentioned in this study, their dd_v6 accession numbers, NCBI accession numbers (if available), sequence and length.
(XLSX)

## Acknowledgments

The author thanks Phil Newmark for his support in this study, which was conducted in his lab. The author also thanks the Newmark laboratory members, past and present, for their constructive feedback throughout this project. The author especially thanks Dr. John Brubacher for his help in writing the manuscript and for critical discussions. We thank Tracy Chong for planarian cartoons used in this manuscript. We thank Dr. Bret Pearson and Dr. Jeffrey Burrows for sharing the transcript ID for *SmedTV*. We thank the University of Wisconsin Biotechnology Center DNA Sequencing Facility for providing DNA-Sequencing and RNA-Sequencing facilities and services. We thank the UW-Madison flow cytometry lab at the University of Wisconsin Carbone Cancer Center (Support Grant P30 CA014520 and NIH grant 1S10RR025483-01) for their help with flow cytometry and analysis. We thank BGU's Bioinformatic core for their assistance in determining if editing events occur in dsRNA structures.

## Author Contributions

**Conceptualization:** Dan Bar Yaacov.

**Data curation:** Dan Bar Yaacov.

**Formal analysis:** Dan Bar Yaacov.

**Funding acquisition:** Dan Bar Yaacov.

**Investigation:** Dan Bar Yaacov.

**Methodology:** Dan Bar Yaacov.

**Software:** Dan Bar Yaacov.

**Supervision:** Dan Bar Yaacov.

**Validation:** Dan Bar Yaacov.

**Visualization:** Dan Bar Yaacov.

**Writing – original draft:** Dan Bar Yaacov.

**Writing – review & editing:** Dan Bar Yaacov.

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
