## [Decision Letter · Decision Letter 0]

8 Dec 2021

Dear Dr. Bar-Yaacov,

Thank you very much for submitting your manuscript "Functional Analysis of ADARs in Planarians Supports a Bilaterian Ancestral Role in Suppressing Double-Stranded RNA-Response" for consideration at PLOS Pathogens. As with all papers reviewed by the journal, your manuscript was reviewed by members of the editorial board and by several independent reviewers. The reviewers appreciated the attention to an important topic. Based on the reviews, we are likely to accept this manuscript for publication, providing that you modify the manuscript according to the review recommendations.

Three reviewers found your manuscript interesting and the study well executed. Reviewers 2 and 3 raised a question about phenotype of the double-knockout ADAR1/ADAR2 mutant. Furthermore, the Reviewer 2 asked about a possible time dependence of the observed contrasting knockout effects of respectively ADAR1 and ADAR2 in readout assays used in the study as well as possible roles of PRLR2 and PRLR3 in the ADAR1/2 pathways. Three reviewers also identified relatively minor issues requiring your attention. Finally, any reasonable extension of the virology-related part would be appreciated by the readership of PLoS Pathogens.

Sincerely,

Alexander E. Gorbalenya, PhD, DSci

Associate Editor

PLOS Pathogens

Mark Heise

Section Editor

PLOS Pathogens

Kasturi Haldar

Editor-in-Chief

PLOS Pathogens

orcid.org/0000-0001-5065-158X

Michael Malim

Editor-in-Chief

PLOS Pathogens

orcid.org/0000-0002-7699-2064

Three reviewers found your manuscript interesting and the study well executed. Reviewers 2 and 3 raised a question about phenotype of the double-knockout ADAR1/ADAR2 mutant. Furthermore, the Reviewer 2 asked about a possible time dependence of the observed contrasting knockout effects of respectively ADAR1 and ADAR2 in readout assays used in the study as well as possible roles of PRLR2 and PRLR3 in the ADAR1/2 pathways. Three reviewers also identified relatively minor issues requiring your attention. Finally, any reasonable extension of the virology-related part would be appreciated by the readership of PLoS Pathogens.

Reviewer Comments (if any, and for reference):

Reviewer's Responses to Questions

**Part I - Summary**

Reviewer #1: The manuscript “ Functional Analysis of ADARs in Planarians Supports a Bilaterian Ancestral Role in Suppressing Double Stranded RNA-Response” by Dan Bar-Yaacov is the first, as far as a I know, functional study on the ancestral role of ADAR activity, revealing that across Bilaterian, it is a key player in reducing the anti-viral activity against endogenous viral-like molecules. The author use of Planaria as a model organism to study ADAR activity is novel and unique and thus the author had to establish the system from scratch. The functional experiments are convincing and the paper is nicely written in a clear manner.

Reviewer #2: In this manuscript, Dr. Yaakov identifies two adenosine deaminases acting on RNA (ADARs) of the planarian species Schmidtea mediterranea, an invertebrate belonging to the superphylum Spiralia. S. mediterranea possesses two enzymes ADAR1 and ADAR2, which are evolutionary related to the mammalian ADAR1 and ADAR2 enzymes. This is in contrast to invertebrates of the superphylum Ecdysozoa (including C. elegans and D. melanogaster), which only express a single ADAR more closely related to mammalian ADAR2. In addition, the author identifies three planarian RIG-I-like receptor homologs (PRLR1-3), which are involved in sensing of double-stranded RNA (dsRNA) and induce innate immune responses. Using RNA-interference, the author knocks down expression of ADAR1 or ADAR2 in S. mediterranea and shows that both ADAR1 and ADAR2 knockdown induce expression of a set of genes homologous to mammalian innate immunity genes. However, only knockdown of ADAR1 induces a lethal phenotype in S. mediterranea characterized by increased cell death in the organism. Importantly, the lethal phenotype can be rescued by additional knockdown of PRLR1, indicating a conserved interplay between ADARs and PRLR-signaling as seen in mammals between ADAR1 and MDA-5. Moreover, the increased expression of innate immunity genes upon knockdown of ADAR1 led to reduced replication of a dsRNA virus infecting, S. mediterranea tricladivirus (SmedTV). Finally, Dr. Yaakov shows that ADAR2, but not ADAR1, edits S. mediterranea mRNAs.

The data presented are generally convincing and novel. The identification of a lower organism that has an antiviral innate immune system reminiscent of that in mammals, as well as two ADAR enzymes with seemingly different functions, is very intriguing and may allow further studies to understand the complex interactions of ADARs with the innate immune system in this simple organism. However, a few things are puzzling and should be addressed by the author.

Reviewer #3: The manuscript „Functional Analysis of ADARs in Planarians Supports a Bilaterian Ancestral Role in Suppressing double-Stranded RNA-Response” by Dan Bar Yaacov is the first study of Adenosine Deaminase Acting on RNA (ADAR) proteins in planarian flatworms. Via thoroughly designed and carefully executed experiments, the author shows that S. mediterranea ADARs function as suppressors of a dsRNA / anti-viral response and engage in editing of endogenous mRNAs.

Specifically, the author shows via RNAi-mediated knock-downs that ADAR1 is essential for survival of S. mediterranea, but not ADAR2. RNAseq analysis of adar1(RNAi) and adar2RNAi) worms reveals that both regulate the expression of many genes related to viral defence and the response to dsRNA, with a high degree of overlap in the genes suppressed by ADAR1 and ADAR2 (upregulated upon RNAi). The functional significance of the dsRNA response upregulation upon ADAR(RNAi) is demonstrated by the downregulation of the recently described dsRNA virus SmedTV in ADAR(RNAi) worms. Further, the rescue of ADAR1(RNAi) lethality via the concomitant knock-down of a planarian RIG-I-like receptor (prlr1) that the authors identify confirms the deep evolutionary conservation between the ADAR and Rig-like receptors in the basal suppression of the dsRNA response. Finally, the author demonstrates endogenous mRNAs editing by ADAR2, but Interestingly, not by ADAR1.

The manuscript is well written, the experiments are generally of high quality and the discussion doesn't make any undue claims. Overall, the study provides a significant advance in understanding the response mechanisms to viruses and the role of ADARs in recognition of self vs. non-self dsRNA in planaria. Little is currently known about planarian innate immune pathways and viral defence mechanisms, hence the manuscript provides an important foundation for further advances in this research area. In addition, the corroboration of the deep conservation of ADAR functions should be of interest beyond the immediate confines of the planarian research field. Therefore, I consider this manuscript principally suitable for publication in PLOS PATHOGENS.

**Part II – Major Issues: Key Experiments Required for Acceptance**

Reviewer #1: (No Response)

Reviewer #2: 1. The author shows that ADAR1 protects S. mediterranea from "autoimmunity" and is proviral for SmedTV, but it does not seem to have editing activity. In contrast, ADAR2 has editing activity in S. mediterranea transcripts, its knockdown reduces SmedTV replication, but seems not to be associated with protection from autoimmune response (lesions and cell lysis), although it also increases expression of innate immunity genes. However, the knockdown efficiency of ADAR1 is much higher than that of ADAR2 (22% vs. ~50% residual transcript levels). The author should investigate, whether the phenotypes are more comparable in a setting where the knockdowns of ADAR1 and ADAR2 are similarly efficient. Maybe with less efficient knockdown of ADAR1, also the lethal phenotype is reduced. In addition it would be interesting to know whether ADAR2 knockdown, which leads to upregulation of innate immunity genes at a later time point than ADAR1 knockdown, will also induce a lethal phenotype with a time delay in S. mediterranea. Is it possible to observe the organisms for a longer period of time? Finally, what happens in a double-knockdown situation of ADAR1 and ADAR2?

2. PRLR1 rescues the lethal phenotype of ADAR1 knockdown. However, the author has identified three PRLRs. He should also test whether PRLR2 and PRLR3 are also involved in the autoimmune response in S. mediterranea after ADAR1 knockdown. At least from the homology analysis (Fig. S7) it appears as if they have the C-terminal domain known to be important for RIG-I function in mammals, and thus might be relevant. In addition and to strengthen the virology aspect of the study, it would be interesting to establish a role for the PRLRs in infection with SmedTV.

3. ADAR-editing in S. mediterranea transcripts seems to be dependent on ADAR2, whereas ADAR1 seems not to have editing activity despite having a putatively functional deaminase domain. The author should check whether editing occurs in SmedTV RNA and, if so, whether ADAR1 and/or ADAR2 are involved.

Reviewer #3: One of the important unresolved issues is the substantial overlap in genes suppressed by ADAR1 and ADAR2, yet the apparent disparity in lethality and RNAi editing capacity of the two homologues. Double knock-downs of the two ADAR homologues would seem to offer a straight-forward and potentially informative approach to further test the extent of functional redundancy between the two homologues, especially with respect to the rate of onset and end point of the effect (e.g. SmedTV downregulation). In case additional experiments are not feasible, the respective section of the discussion should go into more detail and integrate knowledge from other model organisms on possible redundant roles of ADARs.

**Part III – Minor Issues: Editorial and Data Presentation Modifications**

Reviewer #1: I had the opportunity to review an earlier version of this paper elsewhere before- the current version is well improved and much of my earlier comments were resolved.

Currently I have only few minor comments (none is mandatory) :

1- It can be nice to add more analysis to the RNA editing section of the paper:

a. Why so few editing sites were found (although the signal is very clear) is it possible that the cutoff were too stringent?

b. Does editing sites tend to take place within mobile elements in the Planaria genome?

c. Does editing sites tend to take place within predicted dsRNA structures?

d. Any correlation between editing level across tissues and ADARs expression levels?

Reviewer #2: A few additional minor comments:

4. It is mentioned that the ability of S. mediterranea to regenerate amputated heads or tails was not affected by knockdown of either ADAR1 or ADAR2, when performed before onset of lesion formation (in case of ADAR1 knockdown). Please indicate how much ahead of time these amputation experiments were performed. What happens, when amputation is performed closer to the onset of lesion formation?

5. The tables in the supplementary file S2 would be easier to screen if the author would include an additional column with a common protein or gene abbreviation of the homologs. Some, but not all identified homologs, have abbreviations included in the full names.

6. Is it possible to quantify the frequencies of SmedTV-positive cells in figure 3 by flow cytometry? The results may be more accurate than quantification from fluorescent microcopy images.

Reviewer #3: 1) Figure 1B: While the phenotype of adar1(RNAi) worms is clearly visible as described in the text, the images also suggest that adar1(RNAi) animals are much smaller than control- and adar2(RNAi) worms. The adar1(RNAi) worm that is shown for the mild/moderate phenotype has a length of about 6 mm while the control- and adar2(RNAi) worms are about 11 mm in length. Is this difference representative for the whole adar1(RNAi) cohort and maybe due to impaired growth in adar1(RNAi) worms? If this is the case, please report the phenotype in the text. If not, please change the panel accordingly to show a worm of representative size for each cohort.

2) Lines 217-223 & Figure 3A-B: In the main text, figure 3 and figure legend it is not clear whether SmedTV positive cells were only quantified in the head areas shown in Figure 3A or whether cells in the whole body were quantified. Please clarify the analysis. If only SmedTV positive cells in the head areas were quantified, please give a statement or quantification whether a similar effect is visible in other areas of the body. Burrows et al. 2020 showed that SmedTV positive cells are found across the whole body of S. mediterranea. Since the phenotype of ADAR1(RNAi) is least pronounced in the head region (Figure 1B), the effect of ADAR1(RNAi) on SmedTV positive cells might be higher in other regions of the body. Lines 105-106 & Figure S2A: From my point of view, there is no apparent enrichment visible in the brain for WISH stainings of adar1 and adar2. The signal even seems to be weaker in the anterior part and head, respectively, than in the rest of the body. It also seems like there is an enrichment of the signal in cells surrounding the pharynx. Please explain why you conclude an enrichment in the brain or adjust the description of the stainings in the text.

3) Line 110: I think an “of” is missing: “… , we used RNAi knockdown of gene expression.”

4) Lines 113-114: It should be clear that the morphology of adar2(RNAi) animals is not different from control/wildtype worms. Please change the wording accordingly.

5) Lines 256-264 & Figure 4: Please clarify at which timepoint and after how many RNAi feedings, respectively, the phenotype observations were done. Please also include the more detailed explanation of the feeding schedules (initial feeding of only prlr1 dsRNA or control dsRNA) as written in the methods section lines 442-444, in this results section to make the RNAi treatment immediately clear.

PLOS authors have the option to publish the peer review history of their article (what does this mean?). If published, this will include your full peer review and any attached files.

Reviewer #1: No

Reviewer #2: **Yes: **Christian K. Pfaller

Reviewer #3: No

Figure Files:

Data Requirements:

Reproducibility:

References:

---

## [Editor Report · Decision Letter 1]

6 Jan 2022

Dear Dr. Bar Yaacov,

We are pleased to inform you that your manuscript 'Functional Analysis of ADARs in Planarians Supports a Bilaterian Ancestral Role in Suppressing Double-Stranded RNA-Response' has been provisionally accepted for publication in PLOS Pathogens.

Best regards,

Alexander E. Gorbalenya, PhD, DSci

Section Editor

PLOS Pathogens

Mark Heise

Section Editor

PLOS Pathogens

Kasturi Haldar

Editor-in-Chief

PLOS Pathogens

orcid.org/0000-0001-5065-158X

Michael Malim

Editor-in-Chief

PLOS Pathogens

orcid.org/0000-0002-7699-2064
---

## [Editor Report · Acceptance letter]

12 Jan 2022

Dear Dr. Bar Yaacov,

We are delighted to inform you that your manuscript, "Functional Analysis of ADARs in Planarians Supports a Bilaterian Ancestral Role in Suppressing Double-Stranded RNA-Response," has been formally accepted for publication in PLOS Pathogens.

Best regards,

Kasturi Haldar

Editor-in-Chief

PLOS Pathogens

orcid.org/0000-0001-5065-158X

Michael Malim

Editor-in-Chief

PLOS Pathogens

orcid.org/0000-0002-7699-2064